# Prevalence Estimation, Antimicrobial Susceptibility, and Serotyping of *Salmonella enterica* Recovered from New World Non-Human Primates (*Platyrrhini*), Feed, and Environmental Surfaces from Wildlife Centers in Costa Rica

**DOI:** 10.3390/antibiotics12050844

**Published:** 2023-05-02

**Authors:** Ernesto Rojas-Sánchez, Mauricio Jiménez-Soto, Elias Barquero-Calvo, Francisco Duarte-Martínez, Dixie F. Mollenkopf, Thomas E. Wittum, Lohendy Muñoz-Vargas

**Affiliations:** 1Laboratorio de Salud Pública e Inocuidad de Alimentos, Programa de Investigación en Enfermedades Tropicales (PIET), Escuela de Medicina Veterinaria, Universidad Nacional, Heredia 40104, Costa Rica; 2Hospital de Especies Menores y Silvestres, Escuela de Medicina Veterinaria, Universidad Nacional, Heredia 40104, Costa Rica; 3Laboratorio de Bacteriología, Programa de Investigación en Enfermedades Tropicales (PIET), Escuela de Medicina Veterinaria, Universidad Nacional, Heredia 40104, Costa Rica; 4Laboratorio de Genómica y Biología Molecular, Centro Nacional de Referencia de Inocuidad Microbiológica de Alimentos, Instituto Costarricense de Investigación y Enseñanza en Nutrición y Salud, Cartago 30301, Costa Rica; 5Department of Veterinary Preventive Medicine, The Ohio State University College of Veterinary Medicine, Columbus, OH 43210, USA

**Keywords:** *Salmonella*, primates, antimicrobial resistance, Costa Rica, environmental, wildlife, ciprofloxacin, nitrofurantoin

## Abstract

Concern about zoonoses and wildlife has increased. Few studies described the role of wild mammals and environments in the epidemiology of *Salmonella*. Antimicrobial resistance is a growing problem associated with *Salmonella* that threatens global health, food security, the economy, and development in the 21st century. The aim of this study is to estimate the prevalence and identify antibiotic susceptibility profiles and serotypes of non-typhoidal *Salmonella enterica* recovered from non-human primate feces, feed offered, and surfaces in wildlife centers in Costa Rica. A total of 180 fecal samples, 133 environmental, and 43 feed samples from 10 wildlife centers were evaluated. We recovered *Salmonella* from 13.9% of feces samples, 11.3% of environmental, and 2.3% of feed samples. Non-susceptibility profiles included six isolates from feces (14.6%): four non-susceptible isolates (9.8%) to ciprofloxacin, one (2.4%) to nitrofurantoin, and one to both ciprofloxacin and nitrofurantoin (2.4%). Regarding the environmental samples, one profile was non-susceptible to ciprofloxacin (2.4%) and two to nitrofurantoin (4.8%). The serotypes identified included Typhimurium/I4,[5],12:i:-, *S*. Braenderup/Ohio, *S*. Newport, *S*. Anatum/Saintpaul, and *S*. Westhampton. The epidemiological surveillance of *Salmonella* and antimicrobial resistance can serve in the creation of strategies for the prevention of the disease and its dissemination throughout the One Health approach.

## 1. Introduction

In Costa Rica (CR), protected terrestrial areas represent 25.44% of the 51,110 km^2^ territory [1]. The country is recognized worldwide for its biodiversity, with non-human primates (NHPs) being one of the main attractions. The common practice of people, residents, and tourists having close proximity and direct contact with wildlife is considered a risk factor for pathogen transmission [2]. Four NHP species live in the country: spider monkeys (*Ateles geoffroyi*), Central American squirrel monkeys (*Saimiri oerstedii*), howler monkeys (*Alouatta palliata*), and white-faced capuchin monkeys (*Cebus imitator*). According to the International Union for the Conservation of Nature (IUCN), spider and squirrel monkeys are considered endangered, while howlers and capuchins are in the category of vulnerable [3]. Injured wildlife are usually taken to private rescue centers distributed throughout Costa Rica. Wildlife attended in rescue centers with a medical, physical, or behavioral condition that does not allow them to return to the wild can be kept in zoos, sanctuaries, rescue centers, or wildlife exhibits with the corresponding government permits. Among the leading causes of NHP income to these sites are electrocutions, car hits, dog and cat attacks, destruction and fragmentation of habitats, hunting, extensive agriculture, and seizures for illegal possession of wildlife [4].

Concern about zoonoses and wildlife has increased due to the growing proximity of wildlife to humans as a result of human expansion, globalization, climate change, and alterations to the ecosystem [5]. Wildlife can serve as a reservoir of zoonoses, constituting a challenge for public health [6]. Most emerging and re-emerging infectious diseases are zoonotic and include some type of wild reservoir [5,7]. There is a high potential for zoonoses and anthropozoonoses between humans and NHPs [8,9,10,11]. Few studies described the role of wild mammals and environments in the epidemiology of *Salmonella*. This is important since direct exposure to *Salmonella* carriers represents a health risk in ex situ places handling and caring for wildlife [12,13]. *Salmonella* is a Gram-negative, rod-shaped enterobacteria, distributed worldwide, and responsible for causing disease in humans and animals [14,15]. Its extensive distribution, ability to survive under multiple adverse environmental conditions, wide range of hosts, and multiple transmission routes are characteristics that favor infection [12,13,16]. *Salmonella* often occurs asymptomatically in animals [17]; however, clinical signs include acute or chronic diarrhea, enteritis, septicemia, and even abortions [18]. Changes in the eco-epidemiology of *Salmonella* and zoonoses in wildlife have been classified as natural or anthropogenic [19], including human expansion, changes in natural habitats, changes in agricultural activities, globalization of trade, translocation of wildlife, bushmeat markets, consumption of exotic foods, development of ecotourism, access to zoos that allow contact with animals, and possession of wild pets at domestic level [20]. *Salmonella* prevalence in wildlife is variable and not well known. Some studies have reported its isolation from centers dedicated to wildlife management under human care, including serovars with public health importance such as *S.* Typhimurium, I4,[5],12:i:- monophasic variant, Newport, Montevideo, Kentucky, and Heidelberg, among others [14,15,21,22,23,24,25,26]. Studies suggest that the main source of transmission to these animals occurs through the consumption of contaminated water, feed, and surfaces that come into contact with other reservoirs [21,22,25]. Animals under human care can serve as asymptomatic carriers intermittently excreting *Salmonella*. Clinical signs often appear in conjunction with stressful processes, causing a fatal fulminant infection and excreting a large number of bacteria during the course of the disease [11,26,27]. Personnel who work at wildlife centers, visitors, and other animals in the facilities are populations at risk of contagion [14,15,26]. Epidemiological data related to clinical cases and transmission of diseases through indirect contact with animals is limited and usually not considered in the study and approach to infectious disease outbreaks [28,29].

Additionally, antimicrobial resistance is a growing complex health issue associated with *Salmonella* that threatens global health, food security, the economy, and development in the 21st century [30]. This is a natural phenomenon that is constantly expanding and requires a One Health approach to fight [30,31]. In CR, *Salmonella* clinical isolates are mostly pan-susceptible; however, multi-resistant strains of veterinary origin circulate with decreased sensitivity to ciprofloxacin and nalidixic acid [32,33]. Naturally, wildlife does not come into undue contact with antimicrobials [34,35]; however, proximity to urban areas and contact with polluted environments is an important source of resistant bacteria that could be transferred to wildlife populations. Wildlife colonized with antimicrobial-resistant bacteria can act as reservoirs, vectors, bioindicators of antibacterial resistance, and genetic determinants of antibacterial resistance in the environment [34], representing a health risk factor per se [36]. Wildlife under human care can harbor resistant genes and microorganisms due to the dissemination of bacteria and antibiotic wastes in the environment [12,23,25]. However, no previous research related to *Salmonella* and antimicrobial resistance in wildlife under human care in Costa Rica has been performed, including multiple wildlife centers and primate species. Therefore, the aim of this study was to estimate the prevalence and identify antibiotic susceptibility profiles and serotypes of non-typhoidal *Salmonella enterica* recovered from non-human primate feces, feed, and surfaces in wildlife centers in Costa Rica. Understanding the epidemiology of *Salmonella* in these places facilitates the development of control strategies and interventions to protect wildlife conservation and public health. 

## 2. Results

*Salmonella prevalence*. In total, 180 fecal samples from 10 management sites, 133 surfaces, and 43 feed samples were evaluated for *Salmonella enterica*. All *Salmonella* isolates were confirmed by biochemical assays, with a VITEK system and PCR-targeting *hil*A gene, for which all of them were *hil*A positive. From feces, 54 samples belonged to *A. palliata*, 45 to *A. geoffroyi*, 50 to *C. imitator*, and 31 to *S. oerstedii* (Table 1). The estimated prevalence for *A. palliata* was 12.9%, for *A. geoffroyi* 15.5%, for *C. imitator* 16%, and 9.6% for *S. oerstedii*. Prevalences among species were not statistically significant (*p* = 0.852). Environmental samples were distributed in 49 Animal Contact, 45 Human Contact, and 35 Animal–Human Mixed Contact Surfaces (Table 2) with prevalences of 17%, 3.8%, and 10.3%, respectively. Prevalence among surfaces type tended to be significant (*p* = 0.065). Animal Contact and Human Contact prevalences had a significant difference (*p* = 0.027). Eleven samples corresponded to feeds of animal origin, 31 of vegetable origin, and 1 of mixed origin (Table 3) with prevalences of 0%, 3.22%, and 0%. No significant differences were identified among population variables (Table 4). Among the centers willing to participate, we recovered *Salmonella* from feces at 70% (7/10) of the locations, surfaces at 50% (5/10) of the sites, and 10% (1/10) from feed samples. Positive feed samples included a pool of cucumber, tree leaves, and watermelon.

*Antibiotic Susceptibility Profile.* A high percentage of *Salmonella* isolates were considered pansusceptible (78.0% (32/41)), and 22.0% (9/41) were resistant to at least one antibiotic tested. No multi-resistant isolates were recovered in the present study (resistant to at least three families of antibiotics). Resistant profiles included six isolates from feces (14.6%): four non-susceptible isolates (9.8%) to CIP, one (2.4%) to FM, and one to CIP-FM (2.4%). Regarding environments, one profile was non-susceptible to CIP (2.4%), and two to FM (4.8%) (Table 5). One colistin isolated from an environmental sample showed a MIC of 8 μg/mL for colistin. A total of 10 μL of DNA (69.4 ng/mL) from this isolate was sent to The National Bacteriology Reference Center (INCIENSA) to confirm or rule out the presence of the *mcr*-1 gene by PCR. The result was negative. The only isolate from feed was pansusceptible.

*Serotype Characterization.* Of the isolates from feces, 16% (4/25) coincided with serotypes Typhimurium/I4,[5],12:i:-, *S*. Braenderup/Ohio, *S*. Newport, and *S*. Anatum/Saintpaul. *S.* Westhampton was isolated from 13.3% (2/15) of surfaces and 100% (1/1) of feed isolates; 83% (34/41) of isolates obtained a pattern that did not coincide with the reference [38].

## 3. Discussion

Due to zoonotic potential and the threat to wildlife conservation, *Salmonella* has a direct impact on human, animal, and environmental health. Its prevention and control strategies must be addressed from the One Health perspective [29,30,39]. Transdisciplinary work should focus on developing solutions, identifying reservoirs, and reducing cases through biosecurity, personal hygiene, and sanitary food handling practices. In Latin America, there is a lack of information regarding *Salmonella* and NHP, ex situ facilities where they are kept, and the feeds they consume, at least to the authors’ knowledge. In Colombia, *Salmonella* prevalence (80%, *n* = 10) was higher than at any of the participating centers in our study. Authors attribute the high prevalence to a possible outbreak of *Salmonella* at the time of sampling [40]. Other Latin-American groups reported primates suffering from clinical salmonellosis [41,42] and isolates from other wild animals under human care [21,25]. 

One of the largest primate studies at the National Primate Research Center in California reported a prevalence of 12% (*n* = 5076) [43]. This prevalence is similar to our study (13.89%). Primate research centers are sites where hygiene and biosecurity measures are strict and conditions are highly controlled. Likewise, Good et al. suggest that *Salmonella* colonization in primates is rarely associated with clinical symptoms, an observation that coincides with our results (Table 4) in which there is no significant difference in *Salmonella* isolation in relation to abnormal stool consistency (*p* = 0.879). The pressure on animal welfare, better hygiene, and biosecurity conditions has led to the benefit of the management and control of infectious diseases. There was no difference between variables such as life stage, sex, and time under human care for the isolation of *Salmonella* (Table 4). Pathogen prevalence in wildlife may impact the frequency of salmonellosis cases in humans and other animals [23]. *Salmonella* outbreaks in wildlife management sites are infrequent, poorly reported, and unknown in many cases. Visitors and workers pose an increased health risk from direct and indirect contact with asymptomatic animals [22].

*Salmonella* in wildlife management site surfaces is rarely reported. Transmission from surfaces is unknown in many cases and underestimated [12]. The presence of *Salmonella* in feces and environments could suggest environmental contamination from primates. However, environments are open, and the presence of potential vectors such as rodents, birds, insects, fomites, and people were observed, so there may be multiple origins. The environment can serve as a reservoir for *Salmonella*, favoring the colonization of multiple hosts [44]. To determine if the environmental and fecal strains are the same, molecular techniques are needed. However, when wildlife and environments are studied, it is very difficult to conclude in which direction the transmission occurs [45]. In Ohio, the prevalence from environmental samples (10.7%) [12] is similar compared to ours (11.3%). However, among the centers involved in this study, variations in frequencies between 0 and 26% were observed (Table 2), which could be attributed to different environmental conditions, the presence of vectors, hygiene, and biosafety measures. In this study, these characteristics were not included as analysis variables. The prevalence of *Salmonella* on animal contact and human–animal interface surfaces was significantly higher than on surfaces restricted to human contact where cleaning measures are more stringent (Table 2). This could influence the results, as there was a higher frequency of *Salmonella* in animal contact spaces than in areas restricted to human contact (*p* = 0.027).

Feeds included fruits, vegetables, eggs, chicken meat, beef, insects, dry dog feed, wet cat feed, and dairy products; all have been described as sources of *Salmonella* [46,47]. Feeds for wildlife are rarely microbiologically tested even though gastrointestinal disturbances occur frequently [48] and play a key role in transmission due to poor handling and inadequate hygiene measures. In Ohio, the prevalence of *Salmonella* in feed was 15.8% (6/38) [12], which is higher than our study. In India, 3.8% (1/26) of feeds tested were positive, similar to ours. In Mexico, 22 feed samples offered to wildlife were evaluated, all of them negatively [25] coinciding with the result obtained from most centers (9/10) (Table 3). *Salmonella* sampling in feed is important since it can be a source of transmission for wildlife in captivity. Concern has grown around *Salmonella* due to the increased reports of disease outbreaks associated with feed [49]. One of the described mechanisms that favor transmission is that pathogenic bacteria use their fimbriae and cellulose to attach to the surface of plants, which allows them to remain on these feeds [50]. Although quantification of *Salmonella* was not performed on the positive feed samples, it is described that a very low quantity of microorganisms (1 × 10^1^–1 × 10^5^) is required to cause disease [51]. Even in the multiple feed sources included in the composite samples, the *Salmonella* prevalence obtained was relatively low.

Environmental microbiological contamination represents a current challenge, as it is a means of contamination for feed offered to animals [50]. It was common for a primate to take freshly served feed, not finish it, and another primate (or the same) consume it later. This behavior occurs mainly in captivity, where there is limited space, and feeding and deposition of their excreta occur in the same area. This can favor the transmission of diseases between individuals in the same enclosure if an animal is brought into the enclosure and no stool tests are performed to determine its status.

In our study, 78.0% (32/41) of isolates were pansusceptible, while the remaining 22.2% (9/41) were resistant to at least one antibiotic; no multiresistance profiles were found. This matches with previous wildlife studies where the frequency of antibiotic use is low compared to livestock and healthcare settings [13]. Although wildlife is unlikely to be treated with antibiotic therapy, when they enter one of these rescue sites, it is mostly because they have a health condition, which may require the use of antibiotics [52]. The prevalence of resistant profiles found is still significant and matches with resistant profiles found in wildlife and human isolates in Costa Rica. 

The antibiotic with the highest number of resistant isolates was ciprofloxacin (CIP) (Table 5), which, problematically, is one of the first treatment options for salmonellosis in humans [53]. The U.S. National Antimicrobial Resistance Monitoring System (NARMS) and the European Committee for Antimicrobial Susceptibility Assessment (EUCAST) reported an increase in non-susceptible *Salmonella* isolates [54,55]. WHO assigned it as one of the high-priority pathogens for research [56]. In CR, fluoroquinolones such as enrofloxacin are widely used in veterinary medicine as first-line treatment options and prophylactically in primates and domestic animals. Likewise, worldwide, there is high use of fluoroquinolones, including ciprofloxacin, although it is not a first-line choice [57]. Non-susceptible *Salmonella* strains to fluoroquinolones have also been isolated from farm animals [58,59,60,61] as well as raccoons [32] in CR. The surveillance report from CR during the study period reports 7.7% (19/248) of *Salmonella* isolates as not susceptible to ciprofloxacin [62]. The increase in these isolates requires an approach from the One Health perspective to monitor where they are being created and how they spread and determine contagion niches [56].

Nitrofurantoin has been used in veterinary medicine mainly as a growth promoter in livestock. It is banned for growth promotion in Europe [63], as well as in CR. In CR, this drug is registered for use in humans [64]. The Veterinary Drug System does not contain any record of the molecule. Off-label use of this drug in Costa Rica occurs in the veterinary field. In Europe, despite its long-standing ban, nitrofurantoin resistance in *Salmonella* is being maintained over time, mainly by the genes *nfsA* and *nfsB* [63]. Two presumed scenarios that could explain resistant isolates that come from animals with no previous history of using this type of antibiotic are that resistance has been maintained over time in environments and spread through water, feed, or the environment. Another possibility is that it has passed from humans or another animal carrier, through direct or indirect contact. However, establishing the timing of the infection and dissemination of *Salmonella* resistant to nitrofurantoin in these scenarios remains to be studied in greater depth in future studies.

The result of the PCR for the identification of the *mcr*-1 gene was negative, which rules out this mechanism. Different *mcr* genes may be involved in the resistance mechanism. According to INCIENSA, it has been difficult to obtain positive controls for other *mcr* genes. Other mechanisms of colistin resistance in Gram-negative bacteria include chromosomal mutations [65]. The confirmation of resistance to colistin, regardless of the mechanism of resistance to this antibiotic, is through phenotypic methods such as microdilution in broth, a reference methodology recommended by CLSI [37]. 

In CR, clinical *Salmonella* isolates are mostly pan-susceptible; however, multi-resistant strains circulate with decreased sensitivity to CIP. A similar situation occurs in isolates of veterinary origin [32,33,57,58,59,60,61]. These data agree with the findings found in our study, where 14.6% of the isolates have decreased sensitivity to ciprofloxacin. It should be noted that the selection of antibiotics focused on determining the antibiotic susceptibility to infections by *Salmonella* spp. in possible cases of infection in people. Furthermore, since there are no reference MIC values for primates, using the One Health approach to establish the responsible use of antibiotics, hygiene, and biosafety measures could slow antimicrobial resistance due to the selection pressure exerted by antibiotics on bacteria, as well as due to the horizontal transfer of mobile genetic elements.

All the serotypes identified have been associated with human clinical conditions, causing disease outbreaks and contaminating feed. *Salmonella* Typhimurium and its monophasic variant I4,[5],12:i:-represent the first- and second-most prevalent serotypes identified in CR [33,61], with similar situations in Europe [66], USA [67], and Australia [68]. The monophasic variant I4,[5],12:i:-is strongly associated with the feed industry [66,69]. These serovars are generating increasing concern due to the increase in multi-resistant isolates that compromise global health, increasing morbidity and mortality [70]. In CR, isolates with extended-spectrum B-lactamases (ESBL) are already reported [33], which represents a threat to Costa Rican and global public health. *S*. Typhimurium has been identified in primates, causing diarrhea [71] and asymptomatically [27]. It has also been isolated from wildlife in captivity [12,13,24,26] as well as the caretakers of the collection [13]. Variant I4,[5],12:i:- has been reported in captive animals without causing disease [24,26]. To the author’s knowledge, this serovar has not been reported in wild animals in Costa Rica. *S.* Westhampton is a rare serotype [72,73,74]. In CR, this serotype has not been reported [33]. It has been mainly associated with environments where there are animals [72,73,74]. In our study, the samples corresponding to this serotype coincide with what was reported when found in environments and feeds, all belonging to the same management site (Table 2 and Table 3). At this site, the samples from primates do not belong to this serotype, which suggests that the *S.* Westhampton reservoir may be the environment or an unidentified animal. An isolate from feces shared a pattern with two serovars: Braenderup and Ohio, which are indistinguishable by this method [38]. In CR, two Braenderup cases of human origin in 2018 were reported [62]. Worldwide there is a wide description of outbreaks associated with *S.* Braenderup from foods [75,76,77,78,79] and also reports of this serovar in wild birds in captivity [23,26,80]. This serovar has already been isolated from primates [41]. *S*. Ohio is a rare serovar usually acquired from food [81]. However, its reservoirs and routes of transmission through food are often unknown and difficult to establish [81,82]. In CR, this serovar has not been reported according to the reference center [62]. In California, *S.* Ohio has been reported in captive wild birds [26]. Regarding the Newport serotype, in CR, it occupies the eighth place [62]. In recent years, multi-resistant isolates of S. Newport have increased, which has generated concern about the spread of these strains [83,84]. *S*. Newport has been isolated from foods that have caused outbreaks [83,85,86,87], as well as from wild environments and animals [26,83], including primates [41]. *S*. Anatum has been isolated from foods [88,89,90,91] associated with improper handling [90,92]. It also behaves as a causal agent of nosocomial disease [93], which would represent a risk for conservation and collection if adequate biosecurity measures are not in place. According to the CR *Salmonella* report, a case of unknown origin is recorded [62]. *S*. Saintpaul has been isolated from captive wildlife [26]. For CR, two human cases are reported, one from Heredia and another with unknown origin [62]. The Anatum and Saintpaul serotypes are indistinguishable according to the methodology used [38]. Two primates from this collection (Table 4) were positive for the Newport and Anatum/Saintpaul serotypes. The isolation of these serotypes was unique, suggesting that the animals possibly acquired the bacteria at different times. These animals did not share an enclosure.

The PCR methodology was a limitation in identifying the serotypes from this study. More specialized techniques, such as Whole Genome Sequencing or conventional serotyping, are necessary to identify these serotypes. A cross-sectional observational sampling was carried out, so there could be an underestimation of the prevalence if, at that time, the animal(s) did not excrete the microorganism. The study did not intend to determine the causes of colonization, but rather to estimate the prevalence of *Salmonella* from different matrices and their characterization of resistance and serotype. When apparently healthy populations are sampled, there are often false negatives due to intermittent shedding and low shedding load. The effect of these two previously described factors can be counteracted by using feces (as was done) in the culture instead of rectal swabs. Another important aspect that could reduce these false negatives is serial sampling (at least three days) [94]. However, due to the logistics discussed with the management sites and previous experiences in the collection of non-invasive samples, this could not have occurred with all the primates. Further research directions include whole genome sequencing of the isolates, *Salmonella* studies in situ wildlife populations, longitudinal research studies, and risk factor assessment for the isolation of *Salmonella* in the three study matrices.

## 4. Materials and Methods

This cross-sectional study aimed to estimate the prevalence of *Salmonella enterica* in the feces of New World non-human primates under human care in Costa Rica, feed offered to them, and surrounding surfaces that they inhabit at wildlife centers. Centers officially registered at the Ministry of Environment and Energy and willing to participate were included in the study. Sample size (*n*) was obtained using the WinEpi program [95] considering a population of 250 (N) primates [96], an expected prevalence of *Salmonella* spp. in feces of 5%, a confidence level of 97.5% and a margin of error of 2%, resulting in a total of 180 primates. The inclusion criteria considered apparently healthy primates from ten centers, free of any antibiotic therapy during the last week. From those primates, fresh fecal samples, feed (animal and vegetable sources), and three types of surfaces (human contact, animal contact, and human–animal mixed contact surfaces) were sampled in each involved center.

### 4.1. Sample Collection

*Stool*. Approximately 4 g of fresh feces was collected from primates restricted in previously cleaned containment traps. Each primate was expected to defecate there, and the fecal sample was immediately taken aseptically from the top using a sterile spatula contained in the lid of a sterile collection jar (Nipro Medical, Bridgewater, NJ, USA). The samples were identified, stored, and transported at 4 °C to the laboratory.

*Contact surfaces*. Surface samples were aseptically retrieved using personal protective equipment. Sterile gauze pads (Ambiderm, Heredia, Costa Rica) soaked in 10 mL of sterile buffered peptone water (BPW) (Difco^®^, Le Pont de Claix, France) were dragged over the study surfaces and individually placed in sterile bags (Whirl-Pak^®^, Madison, WI, USA). All samples were transported as described above.

*Feeds*. Prepared feeds provided to primates were aseptically collected in sterile bags (Whirl-Pak^®^, Madison, WI, USA) and transported as described above.

### 4.2. Sample Processing

*Salmonella isolation from fecal samples*. A previously standardized culture protocol was performed for *Salmonella* identification [97]. Four grams of feces was homogenized and enriched in 36 mL of Tetrathionate broth (TTB) (BD Co., Spark, MD, USA) to which iodine was added in a ratio of 1:20, followed by incubation for 18–24 h in a water bath at 37 °C. Following, 0.1 mL of the inoculum was placed in 10 mL of Rappaport-Vassiliadis (RV) broth (BD Co., Spark, MD, USA) and incubated in a water bath at 42°C. After 24 h, inoculum was plated on Xylose-Lysine-Tergitol-4 (XLT-4) agar (Remel, Lenexa, KS, USA) and incubated for 18–24 h at 37 °C. *S.* Abaetetuba (ATCC 35640), H2S(+) was used as a positive control, *S*. Cholerasuis (ATCC 10708), H2S(−), and *Escherichia coli* (ATCC 25922) were used as a negative control. A single colony per plate, black or black in the center with a yellow periphery, compatible with *Salmonella*, was transferred to MacConkey (Mck) agar (BD Co., Spark, MD, USA).

*Salmonella isolation from environment and feed*. A total of 100 mL of BPW was added to each sterile bag containing gauze pads and then incubated for 24 h at 37 °C. Then, 1 mL was inoculated to RV broth and incubated in a water bath at 42 °C for 24 h to later be inoculated in XLT-4 and incubated for 24 h at 37 °C. A negative control (sterile gauze) and a positive control (sample inoculated with the *Salmonella* control strain) were included. A single colony per plate, black or black in the center with yellow periphery, compatible with *Salmonella*, was transferred to Mck agar (BD Co., Spark, MD, USA).

*Phenotypic Confirmation.* Lactose-negative colonies with morphologic characteristics compatible with *Salmonella* were placed in test tubes containing 5 mL of Triple-Iron-Sugar agar (TSI) (BD™, Le Pont de Claix, France) and Lysine-Sugar Iron Agar (LIA) (Sigma-Aldrich, St. Louis, MO, USA), and incubated at 37 °C for 24 h. Samples that generate a positive reaction in TSI agar, positive reaction to LIA, and visual glass agglutination test using a polyvalent *Salmonella* antiserum (Denka Seiken Ltd., Tokyo, Japan) were considered as *Salmonella* spp.

*Antibiotic susceptibility test.* The antibiotic susceptibility profiles were obtained using Minimum Inhibitory Concentrations with VITEK^®^2 AST-N279 cards (BioMérieux, Craponne, France). The antibiotics tested were the following: Ampicillin (AM), Ampicillin/Sulbactam (SAM), Piperacillin/Tazobactam (TZP), Cephalothin (CF), Cefotaxime (CTX), Ceftazidime (CAZ), Cefepime (FEP), Imipenem (IPM), Meropenem (MEM), Amikacin (AN), Gentamicin (GM), Nalidixic Acid (NA), Ciprofloxacin (CIP), Nitrofurantoin (FM), Colistin (CL), and Trimethoprim/Sulfamethoxazole (SXT). A bacterial concentration of 0.5 according to the McFarland standard in 3 mL of 0.85% sterile saline (Prelab, San José, Costa Rica) was prepared, followed by a transfer of 145 μL of this solution to 3 mL of sterile 0.45% saline (BioMérieux, France). This second solution was used to obtain the susceptibility profile. Resistance profiles were obtained according to the Clinical and Laboratory Standards Institute [37]_._ Isolates were classified by their Minimum Inhibitory Concentration (MIC) as susceptible and non-susceptible. Intermediate breakpoints were interpreted as non-susceptible. 

*Genotypic confirmation by hilA gene.* All *Salmonella* isolates were tested by PCR reaction targeting the *hilA* promoter gene located in *Salmonella* pathogenicity island I [98]. DNA extraction, purification, and quantification were performed according to a standard protocol [99]. DNA was obtained using a DNeasy Blood and Tissue Extraction Kit (QIAGEN, Hilden, Germany). PCR primers were synthesized using the following sequences: 3′-AGCGTATWGATAATAATCCGGGAT-5′ and 5′-RTTCCACATTTTCTCGGCAATAG-3′ (88 bp). The reactions consisted of a final volume of 22 µL containing 2 µL of the extracted and purified DNA, 10 µL of Master Mix Platinum™ Hot Start PCR (Thermo Fisher, Waltham, MA, USA), 1 µL of primer F, 1 µL of primer R, 4 µL of enhancer (Thermo Fisher, Waltham, MA, USA), and 6 µL of H_2_O (Thermo Fisher, Waltham, MA, USA). The thermocycling conditions included: 1 initial denaturation cycle of 94 °C/15 min, 45 cycles for denaturation and alignment of 95 °C/1 min, 55 °C/1 min, 72 °C/1 min, followed by an extension temperature of 72 °C/10 min. This amplification protocol was carried out in a SimpliAmp thermocycler (Applied Biosystems, Foster City, CA, USA), followed by visualization of the amplicons using the QIAxcel Advanced System technology (QIAGEN, Hilden, Germany). Bands compatible with the expected molecular weights of 88 bp were identified and classified as positive. S. Typhimurium reference strain was used as positive control [100].

*Serotype Identification of Recovered Isolates.* A PCR protocol previously described was carried out for serotype identification [38,101]. Briefly, two multiplex PCR with five primer pairs and one multiplex PCR with two primer pairs were performed. Reactions consisted of a final volume of 25 µL containing 2 µL of extracted and purified DNA, 0.2 mM deoxynucleoside triphosphate, 2 mM MgCl_2_, 5 mM of each primer, and 3.5 units of Taq polymerase contained in Master Mix Platinum™ Hot Start PCR (Thermo Fisher, Waltham, MA, USA). The thermocycling conditions were the same for all reactions: 1 cycle of 94 °C/5 min, followed by 40 cycles of 94 °C/30 s, 56 °C/30 s, 72 °C/1 min, and a temperature extension at 72 °C/5 min, carried out in the SimpliAmp thermocycler (Applied Biosystems, Foster City, CA, USA). The amplicons were observed under the technology of the QIAxcel Advanced System (QIAGEN, Hilden, Germany).

*Statistical Analysis.* Descriptive statistical analysis of *Salmonella* prevalence in feces, feed, surfaces, and susceptibility profiles, was performed. Through a logistic regression analysis, the significance of the *Salmonella* presence (dichotomous response: positive or negative) in feces was evaluated using ANOVA test, considering variables such as primate species, years under human care, age, sex, presence of diarrhea, feces consistency, species, and management center using Minitab19^®^ [102]. Age was determined according to morphological characteristics and records, following a categorization as infant, juvenile, or adult. Four species were considered: *S. oerstedii*, *C. imitator*, *A. geoffroyi,* and *A. palliata*. Sex included female and male. Variables related to feces included diarrhea presence or absence with diarrhea defined as liquid or loose stools with a frequency of three or more times per day and stool consistency defined as normal if stools were firm with a semi-solid consistency and a greenish-brown color and abnormal if stools deviated from the normal description. Time under human care was sorted into four categories: ≤1 year, >1 year–≤5 years, >5 years–≤ 10 years, and >10 years. *p*-values of <0.05 were considered significant in all models. 

## 5. Conclusions

Pathogen transmission between humans and non-human primates is arguably one of the most dangerous outcomes of human–wildlife interactions. Based on the results of this study, we can conclude that captive non-human primates in Costa Rica can excrete *Salmonella* asymptomatically. The presence of *Salmonella* on surfaces in primate environments is an important finding for public health, mainly for people in contact with these primates due to their work duties. The epidemiological surveillance of *Salmonella* and other zoonotic agents at these sites and in wild animals, as well as the patterns of transmission, can favor the creation of strategies for the prevention of the disease and its dissemination.

*Salmonella* was isolated from the three study matrices (feces–environment–feed), and a higher prevalence than expected, based on previous studies, was observed. Significant differences in prevalence were observed between Animal Contact and Human Contact samples. Among recovered isolates, the frequency of ciprofloxacin and nitrofurantoin resistance was considerable. However, no multidrug-resistant isolates were identified.

It was possible to identify the serotype of some of the isolates, and those that were identified have been associated with clinical cases in people and some in primates, representing a risk to public health and wildlife conservation. The variety of characteristics among serotypes makes it likely that the main source of transmission to these animals is through the consumption of contaminated water or feed and surfaces that come into contact with rodents, birds, insects, or other reservoirs of the microorganism. The control of *Salmonella* in these places represents a challenge due to the variety of wildlife species living in proximity—not only primates—which can act as asymptomatic carriers and the working dynamics in these wildlife centers. Sanitary feed handling, constant cleaning, disinfection, and application of biosecurity measures are essential for the control of *Salmonella* in this type of population to avoid clinical cases and to prevent its spread between environments, animals, and people.

## Figures and Tables

**Table 1 antibiotics-12-00844-t001:** Frequency of *Salmonella enterica* recovered from non-human primate fecal samples in 10 wildlife centers in Costa Rica.

	Wildlife Center										
Primate Species	1 *	2	3	4 *	5	6	7	8	9 *	10	Total (%)
*A. palliata*	-	-	1/5	2/31	0/1	-	-	1/3	3/13	0/1	7/54 (12.9)
*A. geoffroyi*	2/5	0/1	3/9	-	0/2	0/1	0/6	1/4	0/5	1/12	7/45 (15.5)
*C. imitator*	-	0/4	0/6	-	0/1	0/10	2/9	3/10	0/3	3/7	8/50 (16)
*S. oerstedii*	1/2	0/5	-	-	-	-	-	-	2/24	-	3/31 (9.6)
Total (%)	3/7	0/10	4/20	2/31	0/4	0/11	2/15	5/17	5/45	4/19	25/180
	(42.9)	(0)	(20)	(6.5)	(0)	(0)	(13.3)	(29.7)	(11.1)	(21)	(13.89)

A *p*-value ≤ 0.05 was considered significant in all comparisons. ANOVA test was used for comparisons between species (*p* = 0.855) and centers (*p* = 0.5). * Significant differences were observed between sites 1 and 4 (*p* = 0.024), 1 and 9 (*p* = 0.024), 4 and 9 (*p* = 0.047).

**Table 2 antibiotics-12-00844-t002:** Frequency of *Salmonella enterica* recovered from environmental surfaces in 10 wildlife centers.

	Wildlife Center										
Surface Type	1	2	3	4	5	6	7	8	9	10	Total (% by Group)
1. Animal Contact *	0/2	0/3	1/7	0/4	0/1	0/2	0/4	0/4	4/7	3/7	8/41 (19.5)
2. Human Contact *	0/2	0/5	1/7	0/5	0/4	0/4	0/6	0/2	0/9	1/9	2/53 (3.8)
3. Mixed Contact	1/3	0/4	0/3	0/2	0/4	0/3	0/1	1/4	2/7	1/8	5/39 (12.8)
Total (% Prevalence per site)	1/7	0/12	2/17	0/11	0/9	0/9	0/11	1/10	6/23	5/24	15/133
	(14.3)	(0)	(11.8)	(0)	(0)	(0)	(0)	(10)	(26)	(20.8)	(11.27)

A *p*-value ≤ 0.05 was considered significant in all comparisons. ANOVA test was used for comparisons between surface type (*p* = 0.08) and centers (*p* = 0.09).* Significant difference was observed between Human Contact and Animal Contact (*p* = 0.027).

**Table 3 antibiotics-12-00844-t003:** Frequency of *Salmonella enterica* recovered from primate feed samples collected from CR wildlife centers.

	Wildlife Center										
Feed Source Protein Type	1	2	3	4	5	6	7	8	9	10	Total (% per Protein Type) (*p* = 0.898)
Vegetable	0/2	0/3	1/4	0/2	0/4	0/5	0/3	0/2	0/2	0/4	1/31 (3.22)
Animal Mixed	0/2	0/2	0/1	0/2	-	0/2	0/1	-	0/2 0/1	-	0/11 (0) 0/1 (0)
Total (% by site) (*p* = 0.938)	0/4	0/5	1/5	0/4	0/4	0/7	0/4	0/2	0/5	1/4	1/43
	(0)	(0)	(20)	(0)	(0)	(0)	(0)	(0)	(0)	(0)	(2.32)

**Table 4 antibiotics-12-00844-t004:** Characteristics of the population of 180 non-human primates included in the study.

Variable	Classification	*n*	(%)	Positives (%)	*p*-Value
Sex	Female	91	50.56	12 (48%)	0.967
	Male	89	49.44	13 (52%)	
Life stage	Infant	18	10.00	2 (8%)	
	Youth	20	11.11	1 (4%)	0.972
	Adult	142	78.89	22 (88%)	
Stool consistency	Normal	155	86.11	22 (88%)	0.879
	Abnormal	25	13.89	3 (12%)	
Time under human care	≤1 year	36	20.00	4 (16%)	
	>1 year–≤ 5 year	40	22.23	8 (32%)	0.832
	>5 year–≤ 10 year	53	29.44	8 (32%)	
	>10 year	51	28.33	5 (20%)	

A *p*-value ≤ 0.05 was considered significant in all comparisons using ANOVA test.

**Table 5 antibiotics-12-00844-t005:** Frequency of susceptibility profiles based on minimum inhibitory concentration (MIC) values of 41 *Salmonella* isolates recovered from feces, feed, and environment.

Class	(g/mL)	0.1	0.3	0.6	0.12	0.25	0.5	1	2	4	8	16	20	32	64	76	128	256	512
	Antimicrobial																		
Beta-Lactams	Ampicillin								41										
	Ampicillin sulbactam								41										
	Piperacillin/Tazobactam									41									
	Cefotaxime							41											
	Ceftazidime							41											
	Cefepime							41											
	Imipenem					41													
	Meropenem					41													
Aminoglycosides	Amikacin								39	2									
	Gentamicin							41											
Quinolones	Nalidixic acid								7	31	3								
	Ciprofloxacin					35	6												
Nitrofuran	Nitrofurantoin											16		21	4				
Polymyxin	Colistin						40				1								
Folate Antagonist	Trimethoprim sulfamethoxazole												41						

The number in the table represents the number of isolates according to their MIC for each antibiotic. Vertical black lines represent the cut-off point for the characterization of susceptibility—susceptible to the left and non-susceptible to the right. Green color represents susceptible isolates, and gray represents non-susceptibles. Colistin has no color and no breakpoint line due to missing data [37].

## Data Availability

Not applicable.

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
