# Peer review of "Prevalence Estimation, Antimicrobial Susceptibility, and Serotyping of Salmonella enterica Recovered from New World Non-Human Primates (Platyrrhini), Feed, and Environmental Surfaces from Wildlife Centers in Costa Rica"

_antibiotics, 2023, doi:10.3390/antibiotics12050844_

Round 1

Reviewer 1 Report

This study investigated the prevalence and antibiotic resistance patterns of Salmonella enterica in various sample types, including fecal samples from primates, environmental samples from animal contact, human contact, and animal-human mixed contact surfaces, as well as food samples from animal and vegetable origins. The authors presented results that provide valuable information on the prevalence and antibiotic resistance patterns of Salmonella enterica in diverse sample types. The findings underscore the need for continuous surveillance of Salmonella infections and emphasize the importance of judicious use of antibiotics to mitigate the emergence of antibiotic-resistant strains. However, to gain a more comprehensive understanding of the epidemiology and antibiotic resistance mechanisms of Salmonella, further studies with larger sample sizes and more detailed molecular characterization of Salmonella isolates are warranted. Overall, this study contributes to the understanding of Salmonella infections and antibiotic resistance patterns in the studied population, and the results are of interest to the scientific community working in the field of microbiology and infectious diseases. Further research in this area has the potential to advance our understanding of Salmonella epidemiology and guide strategies for antibiotic stewardship.

Author Response

Authors appreciate the reviewer comments. Further studies are planned and will be conducted to strengthen this research line in the Region to contribute with mitigation strategies in order to prevent Salmonella and AMR dissemination. 

Reviewer 2 Report

The article presents the prevalence and antimicrobial susceptibility of Salmonella enterica strains isolated from wildlife centers for primates in Costa Rica. It is an interesting and important subject for debate, which should be addressed in the One Health context issue, given the possible role of such places in the harboring and transmission of potentially resistant strains. However, there are some things that need to be adjusted before publishing. 

1. The introduction section does not show much about the pathogenic serotypes and importance of hilA gene which was tested in this research but not found. What is the importance for public health if the strains are non-pathogenic and susceptible to most of the antibiotics tested. 

2. The fact that many food products were tested and  one was identified as positive is important. It should be mentioned what type of food tested positive... 

3.  The discussion section is very detailed but not all the time focused on the subject. In my opinion it could be synthetized better not to lose he reader interest. 

4. There is no conclusion section. I think that given the long discussion section and the literature review performed, which is nicely done, a conclusion which sums up the importance of the results is needed. 

The English editing must be improved. For example, 

1. Line 30: The phrase is too long and must be re-sentenced. 

2. Line 40: replace thru with through

3. Line 105:... the dissemination of these bacteria

4. Line 112: The sentence needs comma adding after .... these places...

5. Line 117: My personal opinion is that it is best to replace food samples with feed .... For me food samples are what is given in human consumption...

6. Line 119: The sentence is very chaotic... it is best to be clearer on what the statistic was about...

7. Line 122: The sentence is very ambiguous. Again, what was tested and showed the significant difference? The p value should be added. 

Author Response

We are grateful for the opportunity to revise our work entitled “Prevalence estimation, antimicrobial susceptibility, and serotyping of Salmonella enterica recovered from new world non-human primates (Platyrrhini), feed, and environmental surfaces from wildlife centers in Costa Rica”. The authors have carefully considered the suggestions and comments provided by you, and important changes were made to the manuscript to comply with the standards of the journal. Please find the listed responses to each comment from the reviewers in the table below. Further comments to improve our work, if any, will be highly appreciated

Reviewer 3 Report

The original research paper entitled “Prevalence estimation, antimicrobial susceptibility, and serotyping of Salmonella enterica recovered from new world non-human primates (Platyrrhini), food, and environmental surfaces from wildlife centers in Costa Rica.” is appropriately well-designed, developed, structured and written by Rojas-Sánchez et al. in appropriate English with a clear structure. They determined the antibiotic susceptibility profiles and serotypes of non-typhoidal Salmonella enterica isolated from non-human primate faecal samples, food offered and surfaces in wildlife centres in Costa Rica. The results were interesting; however, there are some major concerns regarding this manuscript.

-        Golden standard for serotyping is the agglutination-based method by using the antisera kit. Researchers must confirm the serotype of some isolates by using the golden standard methods.

-        Analysis of antibiotic resistance according to the class of antibiotics should be implemented in this study.

-        Multi-drug resistant isolates should be determined and analyzed in this study.

-        It is highly recommended to investigate the relationship or correlation between the serotypes and antibiotic resistance profiles.

-        Heat map analysis also is strongly recommended for analysis and illustration of the results of the antibiotic resistance profile in this study.

-        Discussion section should be thoroughly revised after adding the data which has been mentioned above. 

Author Response

(The authors gave the same response as above.)

Round 2

Reviewer 3 Report

All revisions have been addressed.